# Prediction of Fracture Behavior of 6061 Aluminum Alloy Based on GTN Model

**DOI:** 10.3390/ma15093212

**Published:** 2022-04-29

**Authors:** Fengjuan Ding, Tengjiao Hong, Youlin Xu, Xiangdong Jia

**Affiliations:** 1College of Mechanical Engineering, Anhui Science and Technology University, Bengbu 233100, China; dingfengjuan@ahstu.edu.cn (F.D.); hongtengjiao@ahstu.edu.cn (T.H.); 2College of Mechanical and Electronic Engineering, Nanjing Forestry University, Nanjing 210037, China; youlinxu@njfu.edu.cn; 3College of Mechanical Engineering, Yangtze University, Jingzhou 434023, China

**Keywords:** 6061 aluminum alloy, GTN model, fracture, numerical simulation, orthogonal experiments

## Abstract

To determine the Gurson-Tvergaard-Needleman (GTN)damage model parameters of 6061 aluminum alloy after secondary heat treatment, the uniaxial tensile test was carried out on the aluminum alloy circular arc specimen, and the mechanical properties parameters and the load-displacement curve of aluminum alloy tube were obtained. With the help of the finite element reverse method, scanning electron microscope and a orthogonal test method, the GTN damage model parameters (*f*_0_, *f_N_*, *f_C_*, and *f_F_*) were calibrated, and their values were 0.004535, 0.04, 0.1, and 0.2135, respectively. Then the shear specimen and notch specimen were designed to verify the damage model, the results show that the obtained GTN damage model parameters can effectively predict the fracture failure of 6061 aluminum alloy after secondary heat treatment during the tensile process.

## 1. Introduction

Faced with increasingly serious environmental pollution and energy consumption problems, governments around the world have promulgated strict vehicle emission standards, such as China’s National VI emission standard, Europe’s Euro VI emission standard and the US CAFE emission standard [1]. The main ways to achieve energy saving and emission reduction in automobiles are to improve the level of lightweight, improve the efficiency of power transmission, and optimize the aerodynamic parameters. Among them, improving the level of light weight is the most convenient and effective implementation plan. The “Made in China 2025” plan also requires that automobile manufacturers must increase the application of lightweight materials in automobiles to achieve an average vehicle weight reduction of 5~20% target. Aluminum alloys are widely used in the automotive field due to their lightweight, high specific strength, corrosion resistance, good formability, and recyclability. Problems such as local plastic instability, fracture, spring back, and wrinkling occur, which affect the surface quality and dimensional accuracy of auto parts, resulting in increased scrap rates. To improve the application rate of aluminum alloys in the field of automotive lightweight, the mechanical properties of aluminum alloys, potential internal micro-defects, formability, and the use environment and design requirements of auto parts must be considered. Aluminum alloy materials are prone to fracture failure under the action of forming and applied loads. To deeply study the fracture failure behavior of aluminum alloy materials and establish a suitable failure model, it is necessary to study the relationship between stress triaxiality and fracture strain. Therefore, scholars have paid much attention to the influence of stress triaxiality on the ductile fracture of aluminum alloy materials, to establish its fracture failure model, which can better predict the fracture failure behavior of aluminum alloy, and further reduce the loss of aviation and automobile production fields. At present, the Gurson-Tvergaard-Needleman (GTN) damage model is mainly used to describe the damage and fracture behavior of aluminum alloys. Yildiz et al. [2] measured *ε_N_*, *S_N_*, *f*_0_, *f_C_*, *f_F_,* and *f_N_* in the 6061 aluminum alloy GTN damage model under different aging time conditions with the help of experimental methods, and numerically simulated the 6061 aluminum alloy tensile test based on the GTN damage model. The research results show that the GTN model parameters measured through experiments can be used to simulate the tensile deformation of aluminum alloys under different aging time conditions; the precipitated second phase particles are the source of initial voids; the void volume fraction of aluminum alloys increases exponentially with the effective plastic strain increase. Li et al. [3] studied the damage evolution of AA5182-O aluminum alloy rolled sheet by tensile test and finite element method and used the response surface method to determine the damage parameters of the GTN model. The results show that the strain rate has no effect on *f*_0_ and *f_C_*, but has a significant impact on *f_N_* and *f_F_*. Peng et al. [4] investigated the fracture properties of ductile materials by combining small punch testing, the finite-element-analysis aided testing (FAT) method, and the GTN model, when the simulated load-displacement curves of small punch testing specimens accordant with the experimental results, the parameters of the GTN model could be determined. Bergo et al. [5] discussed the effect of the material characteristic length on the ductile damage and fracture behavior and the mesh sensitivity of the results, their research showed that the only difference between the local and non-local GTN models is the introduction of a material characteristic length through a non-local integral that eliminates the mesh dependency. He et al. [6] proposed an improved shear modified GTN model incorporating two independent damage mechanisms for predicting ductile fracture of 6061 aluminum alloy under different stress states by using a Finite Element inverse identification method incorporating the Latin hypercube design, Kriging approximate model and NLPQL optimization method performed in the optimization software ISIGHT. Chen et al. [7] based on machine learning algorithm proposed an efficient parameters identification strategy of GTN damage model of 2024-T3 aluminum alloy, the strategy combined resilient back-propagation neuro network with genetic algorithm and simulations were implemented in terms of ABAQUS/Explicit. YASSINE’s work presented how the Artificial Neural Network approach determined the parameters of the GTN model in a very short period time [8]. Abbassi et al. [9] identified parameters in the GTN model by tensile test, finite element simulation method, and artificial neural network. The research results show that based on the combination of finite element and neural networks could help to characterize the ductile damage and fracture of metal plates, and Computation time is minimal. Sun et al. [10] used the inverse method based on artificial neural networks, combined with fractional factorial design analysis, the material damage parameters (*f*_0_, *f_N_*, *S_N_*, *ε_N_*, *ks*, *f_C_*, *f_F_*) of the shear modified GTN damage model were identified, their results showed that *f_N_*, *ε_N_*, and *ks* were the significant factors that affect the mechanical behavior of specimen in small punch test. Li et al. [11] established a coupled model by the combination of mechanism-based strain gradient plasticity (MSG) and a shear modified GTN damage model and qualitatively described the size effect on two damage parameters, the research suggested that the MSG theory can promote the evolution of shear damage and inhibit the development of microvoids, and the failure mechanism of materials under high/low-stress triaxiality is nucleation, growth, and coalescence of voids and shear-induced slip, respectively. Henseler et al. [12] conducted a quantitative investigation through In-situ SEM tensile tests and digital image correlation strain measurements to determine the ductile damage behavior of twin-roll cast, hot rolled, and annealed AZ31 thin sheet. Xu et al. [13] mainly determined the parameters *S_N_*, *f_N_*, *f_C_*, *f_F_* in the GTN model utilizing tensile tests and finite element reverse method, whereas other parameters such as *f*_0_ were set to 0 and *ε_N_* was considered to be equal to the strain at which necking occurred. Their study showed that through refined FEM (Finite Element Method) with exact material constitutive relations and a careful determination of the GTN damage parameters, the force-displacement response and the fracture behavior could be well predicted and simulated. Wang et al. [14] proposed a material model based on the Johnson-Cook constitutive model and the GTN model, through Ls-DYNA finite element simulation combined with quasi-static tensile test, high-speed tensile test, and drop weight impact test, it was verified that the proposed material model can describe the structural damage behavior well. Kami et al. [15] established an anisotropic GTN damage mechanics model combined with the Hill’48 anisotropic yield criterion and applied it to the study of the forming limit of the AA6016-T4 sheet, the results show that the forming limit curve predicted by the anisotropic GTN model is in better agreement with the experimental results, especially in the biaxial tension zone. Van Erp et al. [16] used the Gurson-Tvergaard-Needleman (GTN) model and the contour integral crack (CIC) model to simulate the damage and failure behavior of a series 5000 aluminum-magnesium alloy during small punch tests. Ying et al. [17] conducted the hot tensile test for AA7075 and fitted the corresponding flow behavior by Hensiel-Spittel (HS) constitutive equation, then the GTN mesoscopic damage model was implemented to accurately characterize the damage evolution phenomenon of the materials at elevated temperatures. The establishment of damage models for aluminum alloy sheets based on different stress states has achieved certain research results and has been widely used in actual production [18,19,20]. However, there are few studies on GTN damage models for aluminum alloy extruded tubes under heat treatment conditions. Therefore, it is necessary to further explore the damage evolution law of aluminum alloy extruded tubes under heat treatment condition, and to improve the forming performance of aluminum alloy extruded tubes. In addition, the parameters in the GTN damage model are difficult to measure by direct methods, and how to accurately obtain the parameters of the GTN model needs further research and improvement [21,22]. At present, the most widely used aluminum alloy materials in industrial production are mainly extruded tubes. Therefore, commercial 6061-T6 aluminum alloy extruded tubes were selected as the research object in the experiment. Through heat treatment test, quasi-static tensile test, and orthogonal test combined with finite element simulation method, the deformation and damage law of 6061 aluminum alloy under different stress triaxiality was studied. The GTN damage model establishes a coupled damage constitutive modeland performs finite element simulation on the tensile deformation of 6061 aluminum alloy specimens. By comparing the tensile simulation results and test results of the GTN damage model on different types of aluminum alloy specimens, the applicable range of the GTN damage model under different stress states is found, and the correctness of the finite element simulation of the damage model is verified.

## 2. Materials and Experimental

The material used in the test is a 6061-T6 aluminum alloy extruded tube(BIG LIGHT GROUP, Suzhou, China) with a thickness of 6 mm and an outer diameter of 90 mm, its general mechanical properties such as the yield strength, tensile strength, and uniform elongation are 256 MPa, 278 MPa, and 11%, respectively. In addition, its chemical composition is shown in Table 1, and the data are provided by the manufacturer.

According to GB/T228-2010 ⟪Metal Material Room Temperature Tensile Test Method⟫, cut the arc specimen, shear specimen, arc notch specimen, and V-notch specimen along the axial direction of the 6061-T6 aluminum alloy tube, as shown in Figure 1, Figure 2, Figure 3 and Figure 4. According to the research results of Ding et al. [23], 6061-T6 aluminum alloy can obtain the best strength and plasticity at the heating temperature of 560 °C, water cooling, and heat preservation for 4 h. Therefore, heat treatment is performed on different types of aluminum alloy specimens (heating temperature was 560 °C, holding time was 4 h, and then were cooled in water). Figure 3 shows the arc-notched specimens with the notch radii of 5 mm, 6 mm, 8 mm, 11 mm, and 15 mm, Figure 4 shows the V-notch specimens, the notch angles are 60°, 90°and 120°, respectively.

To explore the failure characteristics of 6061 aluminum alloy under different stress states after secondary heat treatment, the UTM5105 electronic universal testing machine (XinSansi, Shanghai, China)was used to perform tensile tests on the 6061 aluminum alloy circular arc specimen, shear specimen and notched specimen, and a 50 mm YSJ50/10-ZC extensometer (XinSansi, Shanghai, China)was used to measure the load-displacement curve of the gauge length section of the aluminum alloy specimen. A constant loading speed was used in the test, among which the loading speeds of the arc specimen, the shear specimen, and the notched specimen were 2 mm/min, 0.08 mm/min, and 0.2 mm/min, respectively.

The load F and displacement ∆L curve of 6061 aluminum alloy extruded tube were obtained through the unidirectional tensile test, which was converted into a real stress-strain curve according to Formulas (1) and (2), as shown in Figure 5.
(1)$$\varepsilon_{true}=ln(1+\frac{\Delta l}{l_{0}})=ln(1+\varepsilon_{e})$$
(2)$$\sigma_{true}=\sigma_{e}\left(1+e\right)$$
(3)$$\varepsilon_{pl}=\left|\varepsilon_{true}\right|-\left|\varepsilon_{e}\right|=\left|\varepsilon_{true}\right|-\frac{\left|\sigma_{true}\right|}{E}$$
where *ε_true_* and *σ_true_* are true strain and true stress, respectively; *E* is elastic modulus; *ε_e_* and *σ_e_* are engineering strain and engineering stress, respectively; *ε_pl_* and *ε_el_* are effective plastic strain and elastic strain, respectively.

The strain hardening curve of a material is an important way to characterize its plastic deformation behavior. According to the true-stress-strain curve of the 6061 aluminum alloy tube shown in Figure 5, it can be seen that the flow stress of the 6061 aluminum alloy tube increases as the strain increases. The formula of aluminum alloy hardening law is as follows [24]:(4)$$\sigma_{true}=K\varepsilon_{pl}{}^{n}$$

In Formula (4), *σ_true_* is the true stress; *ε_pl_* is the effective plastic strain, *K* and *n* are the undetermined material hardening constants. The least-square method is used to fit the stress-strain curve of the 6061 aluminum alloy tube to obtain the *K* and *n* values. The calculation results are shown in Figure 6. The mechanical performance parameters of the 6061 aluminum alloy tube are shown in Table 2.

In Table 2, E is Young’s modulus (MPa), A is Uniform elongation (%), σs is yield stress (MPa), σb is tensile stress (MPa), K is strength coefficient and *n* is hardening coefficient.

## 3. GTN Model and Determination of Damage Parameters

### 3.1. Introduction to GTN Model

Tvergaard [25] modified the Gurson model by considering the interaction between the holes and introduced correction coefficients *q*_1_, *q*_2_, and *q*_3_, and adjusted the model to a numerical form. Tvergaard and Needleman [26] introduced the damage function *f^Φ^* to replace the void volume fraction in the original model to characterize the effect of void aggregation when the material ruptures. The GTN model obtained has the following mathematical expression:(5)$$\Phi =({\frac{\sigma_{eq}}{\sigma_{m}})}^{2}+2q_{1}f^{\varphi }cosh(-\frac{3q_{2}\sigma_{h}}{2\sigma_{m}})-1-q_{3}{\left(f^{\varphi }\right)}^{2}=0$$

In Formula (5), *σ_eq_* is the macroscopic Von Mises equivalent stress; *σ_m_* is the equivalent stress of the matrix material; *q*_1_, *q*_2_, and *q*_3_ represent the correction coefficients of the interaction between the pores. When *q*_1_ = *q*_2_ = *q*_3_ = 1, GTN The model degenerates to the original Gurson model; *σ_h_ =* (1/3) *σ_kk_* is the macroscopic hydrostatic stress; *f^ϕ^* is the effective void volume fraction, which is a function of the void volume fraction *f*, and its mathematical expression is as follows [27]:(6)$$f^{\Phi }=\left\{\begin{array}{ll} f, & f\leq f_{c}\\ f_{c}+\frac{\left(1/q_{1}-f_{c}\right)}{f_{F}-f_{c}}(f-f_{c}), & f>f_{c} \end{array}\right.$$

In Formula (6), *f* is the void volume fraction, *f_C_* is the critical void volume fraction; *f_F_* is the critical void volume fractionate failure.

The plastic flow of the porous material is related to the cumulative plastic strain of the matrix material and the void volume fraction *f*. The evolution equation of the equivalent plastic strain of the matrix material can be obtained by the principle of equivalent plastic work:(7)$$\left(1-f\right)\sigma_{m}d{\bar{\varepsilon }}_{m}^{\mathrm{pl}}=\sigma :d\varepsilon^{p}$$

In Formula (7), $d{\bar{\varepsilon }}_{m}^{\mathrm{pl}}$ is the cumulative equivalent plastic strain increment of the matrix material; $d\varepsilon^{p}$ is the equivalent plastic strain increment. The change in the volume fraction of microvoids in the process of plastic deformation of metal materials is mainly composed of two parts: the growth of the original voids and the change in the volume fraction of microvoids caused by the nucleation of new voids. The damage evolution can be expressed as follows:(8)$$df=df_{g}+df_{n}$$
where *df_g_* is the void growth, which can be derived from the mass conservation law, as presented in Formula (9).
(9)$$df_{g}=\left(1-f\right)d\varepsilon^{p}:I$$

In Formula (9), $d\varepsilon^{p}$ is the equivalent plastic strain increment, *I* is a second-order unit tensor. The nucleation process is strain-controlled. The new void nucleation is expressed as the following equations (Formulas (10) and (11)):(10)$$df_{n}=Ad\varepsilon_{m}^{pl}$$
(11)$$A=\frac{f_{N}}{S_{N}\sqrt{2\pi }}exp\left(-\frac{1}{2}(\frac{{\bar{\varepsilon }}_{m}^{pl}-\varepsilon_{N}}{S_{N}}{)}^{2}\right)$$

In Formula (11), *A* is the nucleation coefficient of the void; *f_N_* is the nucleate void volume fraction; *S_N_* is the standard deviation of void nucleation strain; *ε_N_* is the average equivalent plastic strain at nucleation, $d{\bar{\varepsilon }}_{m}^{\mathrm{pl}}$ is the cumulative equivalent plastic strain increment of the matrix material; $d\varepsilon^{p}$ is the equivalent plastic strain increment.

### 3.2. Determination of Damage Parameters of GTN Model

Based on the GTN damage model combined with the finite element method to analyze the damage evolution and failure of 6061 aluminum alloy during the tensile process, the SEM scanning electron microscope analysis and the orthogonal experiment were combined to obtain the damage parameters of the test conditions. Sun et al. [28] proposed to obtain f_0_ by calculating the area fraction of the initial hole. First, observe the distribution of holes in the 6061 aluminum alloy arc specimen at different stretching stages through the JSM7600F field emission scanning electron microscope, to determine the relevant damage parameters of the GTN damage model.

(1)The initial void volume fraction *f*_0_.

The void volume fraction of the 6061 aluminum alloy specimen before plastic deformation is the initial void volume fraction, namely *f*_0_. The initial pores in aluminum alloy materials originate from the accumulation of vacancies and second-phase particles during the manufacturing process. The initial microstructure of the undeformed aluminum alloy material was obtained by scanning electron microscopy, as shown in Figure 7.

(2)Fracture pore volume fraction *f_F_*.

Figure 8 shows the morphology of the 6061 aluminum alloy specimen after a tensile fracture. The fracture is mainly composed of pores of various sizes. After reaching the critical value of the pore volume fraction, the micropores in the material are interconnected.

Based on of the SEM pictures of 6061 aluminum alloy arc specimens in different tensile stages obtained by scanning electron microscope, the image analysis, and measurement software Image-Pro Plus 6.0 is used to calculate the initial stage and fracture of the aluminum alloy arc specimens The average area percentage of pores in different stages is used to determine the volume fraction of pores in different deformation stages. The measurement results are shown in Table 3.

### 3.3. Orthogonal Test

Considering the complex nonlinear relationship between the load-displacement curve obtained by the tensile test of the 6061 aluminum alloy arc specimen after the second heat treatment and the parameters of the GTN damage model, the commonly used finite element reverse calibration method is mainly based on the experimental data, the GTN model parameters were modified repeatedly to minimize the error between the simulated load-displacement curve and the test curve, to determine the most suitable GTN damage parameters. This method of repeated attempts and modifications consumes a lot of time and lacks theoretical support. An orthogonal experimental design method was used to design the test, and with the aid of ABAQUS finite element simulation software, the uniaxial tensile test of 6061 aluminum alloy was numerically simulated to optimize the damage parameters *ε_N_*, *f_N_*, *f_C_*, *f_F_* in the GTN model. According to existing studies, the yield surface correction coefficients *q*_1_ = 1.5, *q*_2_ = 1, *q*_3_^2^ = *q*_1_^2^ [29] in the GTN damage model are in good agreement with the experimental results for most materials, and this research has also been verified by other scholars [30]. Chu and Needleman [31] pointed out that the hole nucleation parameters can be taken as constants, namely *ε_N_* = 0.3, *S_N_* = 0.1 [32]. GULLERUD et al. [33] proposed an empirical value *f_C_* with a value range of (0.1~0.2), and the hole nucleation parameter *f_N_* has no fixed value, and its value range is between 0.005 and 0.04 [34]. Since the parameter values in the GTN damage model are uncertain, it is difficult to measure them by experimental methods. To reduce the calculation cost and the number of experiments, orthogonal experiments are used to optimize the experimental design. To study the influence of damage parameters on the damage behavior of 6061 aluminum alloy circular arc specimens after secondary heat treatment, the damage parameters *ε_N_*, *f_N_*, *f_C_*, and *S_N_* are selected as the factors of the orthogonal test, and each factor is set to 4 levels, thus the orthogonal table of L_16_ (4^4^) is established, the experimental factors and levels are shown in Table 4, and the experimental design scheme is shown in Table 5.

### 3.4. Numerical Simulation Based on GTN Model

In the research, the commercial software ABAQUS/Explicit dynamic explicit module was selected to solve and analyze the whole tensile process of 6061 aluminum alloy tubes. According to the actual size of the 6061 aluminum alloy tensile specimen, a three-dimensional solid finite element model of the tensile specimen was established. The element type adopted was the 8-node reduced integral element C3DR8 considering the hourglass control, as shown in Figure 9. To obtain a mesh with uniform shape and size, it was necessary to divide the created model before dividing the mesh. In addition, starting from the element size of 0.5 mm–5 mm for grid sensitivity analysis, considering the finite element calculation time and simulation accuracy, the global size of the grid element of the arc specimen was set to 5 mm, and the local size is set to 1 mm.

Set the same boundary conditions as the tensile test. One end of the finite element model of the aluminum alloy tensile specimen was fixed, and it was not allowed to move or rotate in any direction, while the two surfaces of the other end were coupled to the reference point through coupling, and the coupling point applies a velocity load of 2 mm/s along the tensile direction. The addition of coupling constraints makes the output of the reaction force in the numerical simulation of stretching more convenient. To ensure that the numerical simulation of stretching was more accurate, in the field output results, check the VVF, state variable STATUS, shear damage, scalar stiffness degradation, and damage initiation criteria. Check the RF in Forces/Reactions and UT in Displacement to output the reaction force of the coupling point and the displacement of the two reference points of the 6061 aluminum alloy specimen gauge section. According to the relative displacement difference between the two reference points of the gauge length section, combined with the reaction force of the coupling point, the load-displacement curve of the aluminum alloy specimen during the tensile process was obtained.

According to the orthogonal test design scheme shown in Table 5, the uniaxial tensile numerical simulation of the 6061 aluminum alloy arc specimen is carried out. The flow stress in the experimental results is directly used in the simulation, and the load-displacement curve of the gauge length section of the aluminum alloy specimen is output in the simulation results, and is converted into the true stress-strain curve according to Formulas (1) and (2). Figure 10 shows the true stress-strain curves of 16 sets of simulation outputs and compares them with the test curves.

As shown in Figure 10, the change of GTN damage parameters has a certain effect on the stress-strain curve, the maximum stress, and the strain corresponding to the maximum stress simulated by the tensile test. In addition, GTN damage parameters also affect the tensile fracture strain and uniform elongation of 6061 aluminum alloy specimens. Therefore, the orthogonal test indicators are: RE_s_ = |simulated true strain-test true strain|/true strain × 100%
RE_σ_ = |simulated true stress-test true stress|/true stress × 100%

To better understand the influence of GTN damage parameters on the mechanical properties of 6061 aluminum alloy, the maximum stress and strain obtained by the simulation are compared with the test results. The analysis of the orthogonal test results is shown in Table 6. *K_ij_* in the table represents the sum of test indicators of the corresponding level *i* in the *j*th column, *i* = 1, 2, 3, 4. *R_j_* represents the range of factors in the *j*th column, *j* = 1, 2, 3, 4. Since *RE_ε_* and *RE_σ_* are smaller, the true stress-strain curve of the aluminum alloy tensile specimen output by the corresponding GTN parameter combination is closer to the experimental curve, therefore in each factor, the corresponding level of min (*K*_1*j*_/4, *K*_2*j*_/4, *K*_3*j*_/4) is the best level.

According to the analysis of the orthogonal test results shown in Table 6, the optimized conditions analyzed by the two indicators, *RE_ε_* and *RE_σ_* are inconsistent. Therefore, it is necessary to comprehensively consider the optimal process conditions according to the priority of the influencing factors. Since the range of the factor *f_N_* is the largest, its influence on *RE_ε_* and *RE_σ_* ranks first, take (*f_N_*)_4_ at this time. For the factor *f_C_*, its influence on *RE_σ_* ranks second, and (*f_C_*)_1_ is acceptable at this time. For the factor *ε_N_*, its influence on *RE_ε_* ranks second, and (*ε_N_*)_4_ is desirable at this time. For the factor *S_N_*, its influence on *RE_ε_* and *RE_σ_* ranks last, but the impact of *S_N_* on *RE_ε_* is greater than that on *RE_σ_*, so take (*S_N_*)_3_ at this time. Therefore, (*ε_N_*)_4_(*f_N_*)_4_(*f_C_*)_1_(*S_N_*)_3_ is the optimal level, that is, *ε_N_* = 0.4, *f_N_* = 0.04, *f_C_* = 0.1, *S_N_* = 0.1. The GTN damage model parameters obtained by the finite element reverse method combined with the orthogonal test are shown in Table 7.

According to Figure 9 and Table 6, it can be seen that the GTN parameters in the simulation models 1, 2, 3, 4, 6, 7, 8, 9, 10, 11, 13, 14, and 15 lead to an increase in the elongation of the 6061 aluminum alloy tensile specimens. The error between the stress-strain curve obtained from the simulation and the test curve is relatively large. Although the stress-strain curves calculated in the simulation models 12 and 16 are in good agreement with the test curves, model 16 is the best for fracture prediction. Compared with the experimental curve, the relative prediction errors of the maximum stress and strain calculated by Model 16 are 5.14% and 1.64%, respectively. Figure 11 is a comparison of the fitted stress-strain curve and experimental curve in model 16.

By comparing the test curve and the simulation curve in Figure 11, it can be seen that the GTN damage parameters in Model 16 can better simulate the tensile deformation of the aluminum alloy specimen, which also shows that the optimal level selected in the orthogonal test is reasonable. Therefore, the parameters listed in Table 7 can be used as the final values of the parameters of the GTN damage model. The ABAQUS software was used to numerically simulate the uniaxial tensile process of 6061 aluminum alloy circular arc specimens, and the equivalent plastic strain cloud diagrams and void volume fraction cloud diagrams at different stretching stages were obtained, as shown in Figure 12 and Figure 13.

As can be seen from Figure 12, from the initial stage of tensile deformation to when the tensile displacement is 9.8 mm, the equivalent plastic strain of the aluminum alloy specimen is 0.1789, and the aluminum alloy specimen is in the uniform plastic deformation stage. When the tensile displacement is 29.26 mm, the equivalent plastic strain of the aluminum alloy specimen is 0.9138. At this time, the aluminum alloy specimen is deformed and concentrated. With the continuous increase of the tensile displacement, the aluminum alloy specimen necks. Then when the tensile displacement reaches 30.45 mm, the equivalent plastic strain of the aluminum alloy specimen is 0.9725, and fracture occurs at the maximum equivalent plastic strain.

Figure 13 shows the cloud diagram of the void volume fraction of the 6061 aluminum alloy at different tensile displacements. From the beginning of the tensile stage to the displacement of 9.8 mm, the void volume fraction of the sample is 0.0006212, and the void volume fraction is relatively small. Small plastic deformation occurs in the interior, the degree of material damage accumulation is small, and the specimen does not fracture. When the tensile displacement of the specimen reaches 29.26 mm, the maximum void volume fraction of the aluminum alloy specimen is 0.09843, which is relatively close to the critical void volume fraction. At this time, the aluminum alloy specimen does not fracture. When the tensile displacement reaches 30.45 mm, the maximum void volume fraction of the aluminum alloy specimen is 0.1134, and the damage accumulates on the central element of the specimen, resulting in the necking phenomenon. As the tensile test continued, the centrally located unit broke rapidly. In the tensile deformation simulation, the void volume fraction increases with the increase of the tensile displacement, which indicates that the increase in the plastic deformation of the material leads to the accumulation of internal damage to the material.

### 3.5. Numerical Simulation Results

The GTN damage model is used to numerically simulate different types of 6061 aluminum alloy specimens, and the load-displacement curve of the gauge length section of the aluminum alloy specimen is output for comparison with the test curve, as shown in Figure 14.

According to the comparison between the simulated load-displacement curve and the test curve of the 6061 aluminum alloy specimen under different stress states shown in Figure 14, the GTN model is more accurate for the simulation results of aluminum alloy arc specimens, arc notch specimens, and V-notch specimens, but when simulating the shear specimen, the simulated curve strain reaches a certain level, and the loading does not decrease. This is consistent with the findings of the literature [35].

The mechanism of GTN is that the material with initial holes nucleates, aggregates, connects, and finally fractures breaks under the action of tensile load. Through numerical simulation of the tensile process of 6061 aluminum alloy specimens under different stress states, the equivalent plastic strain cloud diagram and stress triaxial distribution cloud diagram corresponding to the minimum cross-section of the aluminum alloy specimen is obtained. Since the triaxial stress of the shear specimen is close to 0 during the tensile process, failure to provide the driving force required for hole growth. Therefore, the GTN model is not suitable for the lowstress triaxiality range, and the fracture displacement prediction accuracy of the 6061 aluminum alloy circular arc-notched specimens and V-notched specimens in the highstress triaxial range is high.

## 4. Conclusions

In the study, uniaxial tensile tests were performed on 6061 aluminum alloy specimens under different stress states. The GTN damage model parameters were determined with the help of scanning electron microscopy, finite element analysis, and orthogonal test methods, and different types of aluminum alloy tests were performed based on the GTN damage model. Numerical Simulation. The main conclusions are as follows:

(1) The elastic modulus, yield strength, tensile strength and uniform elongation of the 6061 aluminum alloy tube after secondary heat treatment (the heating temperature is 560 °C, the holding time is 4 h, water-cooling mode) are 65,907 MPa, 152 MPa, 412.29 MPa, and 26%, respectively. Furthermore, the plastic deformation behavior of the 6061 aluminum alloy tube can be well characterized by the Hollomon equation. 

(2) In the initial stage of stretching, there are a small number of small pores on the surface of the 6061 aluminum alloy arc specimen. With the increase of deformation, the number of dimples at the fracture increases, the dimples are elongated into a parabolic shape and the second phase particles can be seen in the dimples. The fracture mechanism of the 6061 aluminum alloy arc specimen is a ductile fracture. 

(3) The damage parameters *f*_0_, *f_N_*, *f_C_*, and *f_F_* of 6061 aluminum alloy GTN model were obtained by image analysis method, finite element method and orthogonal experiment method, and their values were 0.004535, 0.04, 0.1, and 0.2135, respectively. The metallographic numerical simulation of the uniaxial tensile test of the 6061 aluminum alloy arc specimen was carried out by using the GTN model, and the fitted load-displacement curve was very consistent with the test curve. The reliability of the GTN damage model parameters was verified by the arc-notched specimen and the V-notched specimen, and the damage model can effectively predict the load capacity and failure capacity of the 6061 aluminum alloy tube.

## Figures and Tables

**Figure 1 materials-15-03212-f001:**
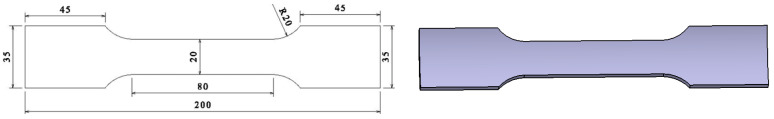
Arc tension specimen.

**Figure 2 materials-15-03212-f002:**
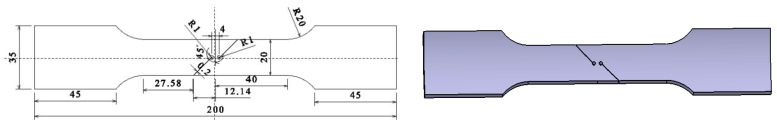
Shear tension specimen.

**Figure 3 materials-15-03212-f003:**
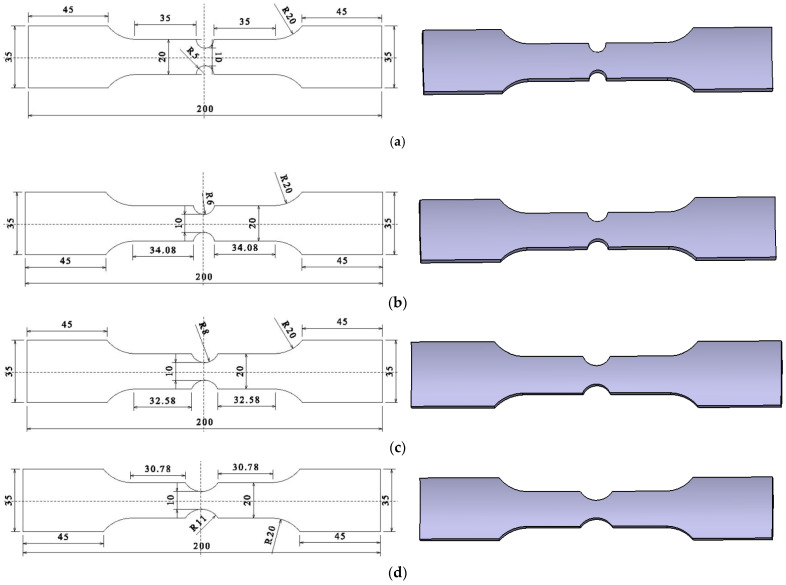
Arc-shaped notched tensile specimen. (**a**) R = 5 mm; (**b**) R = 6 mm; (**c**) R = 8 mm; (**d**) R = 11 mm; (**e**) R = 15 mm.

**Figure 4 materials-15-03212-f004:**
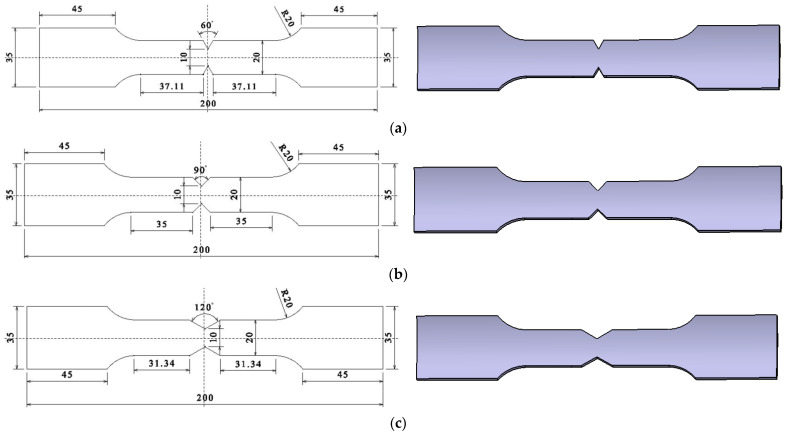
V-shaped notched tensile specimen. (**a**) α = 60°; (**b**) α = 90°; (**c**) α = 120°.

**Figure 5 materials-15-03212-f005:**
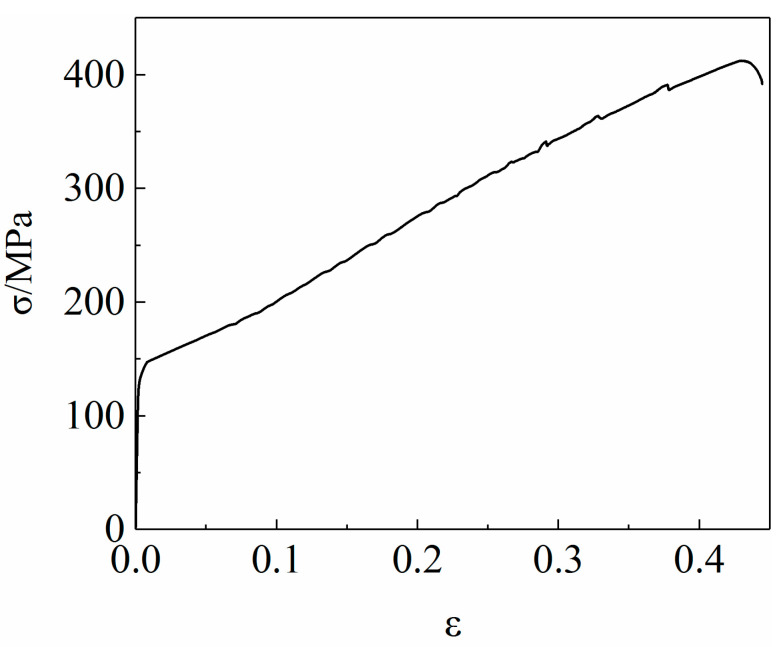
True stress-strain curve of 6061 aluminum alloy tube.

**Figure 6 materials-15-03212-f006:**
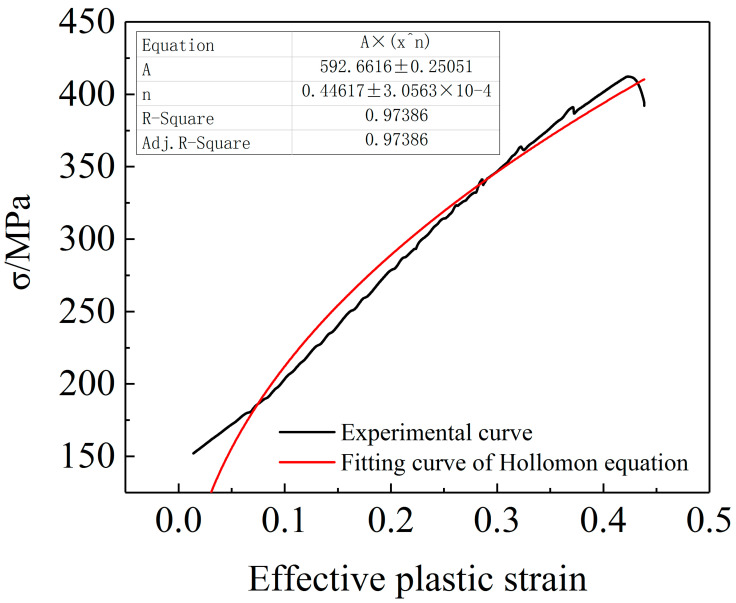
Fitting results of true stress-plastic strain curve.

**Figure 7 materials-15-03212-f007:**
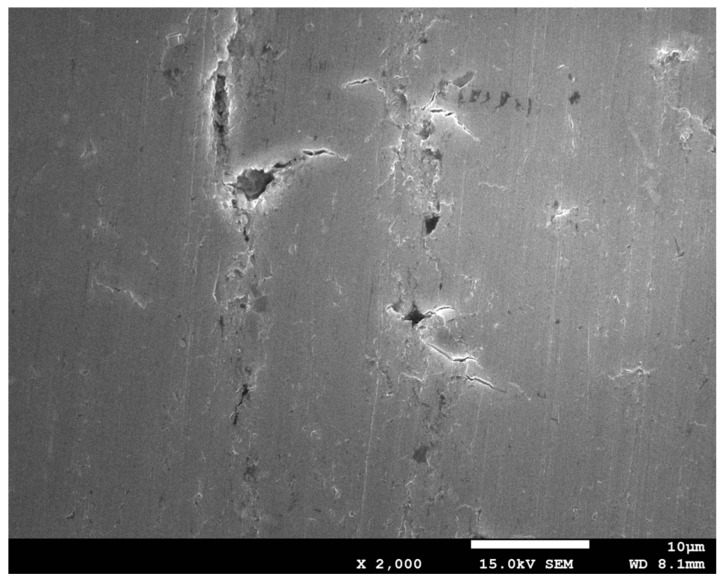
Microstructure of 6061 aluminum alloy specimen without deformation.

**Figure 8 materials-15-03212-f008:**
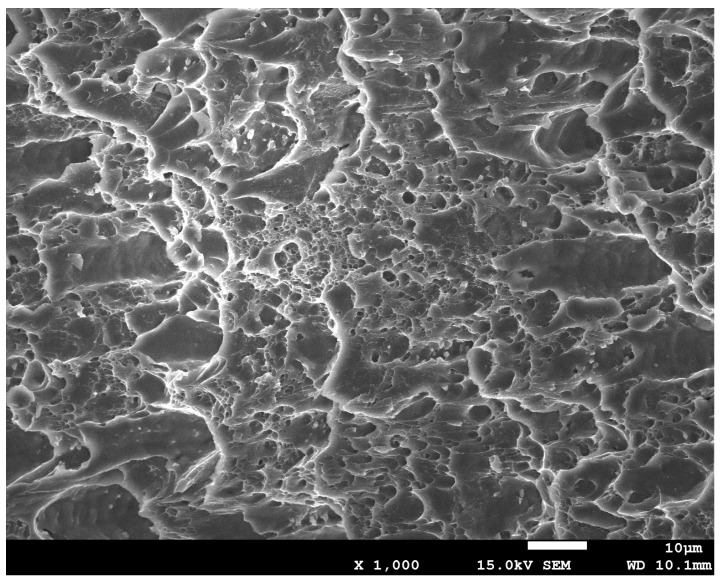
Microstructure of 6061 aluminum alloy after fracture.

**Figure 9 materials-15-03212-f009:**
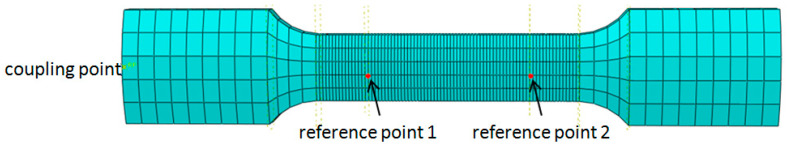
Finite element model of 6061 aluminum alloy tensile specimen.

**Figure 10 materials-15-03212-f010:**
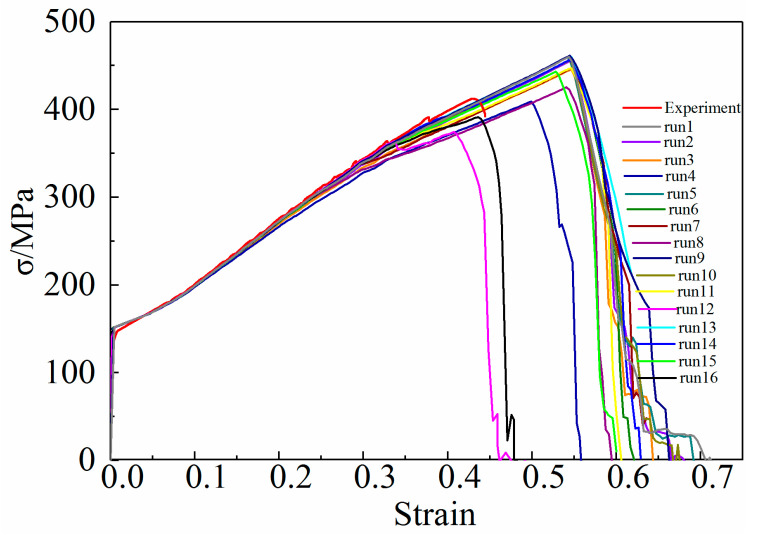
Comparison of the numerical and experimental stress-strain curve.

**Figure 11 materials-15-03212-f011:**
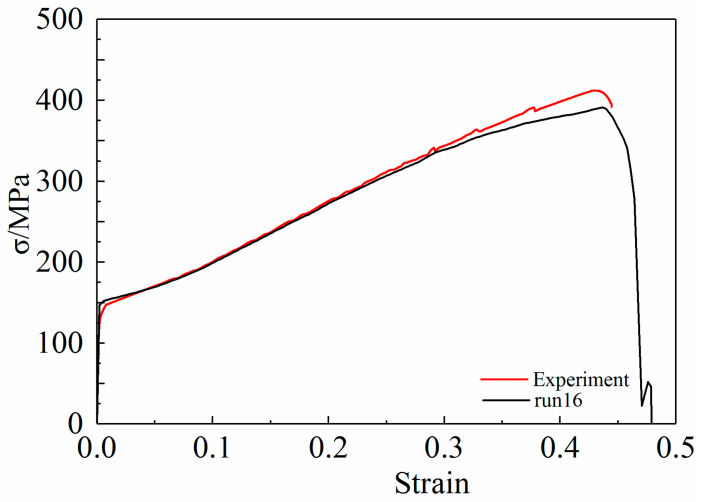
Comparison of run 16 and experimental stress–strain curve.

**Figure 12 materials-15-03212-f012:**
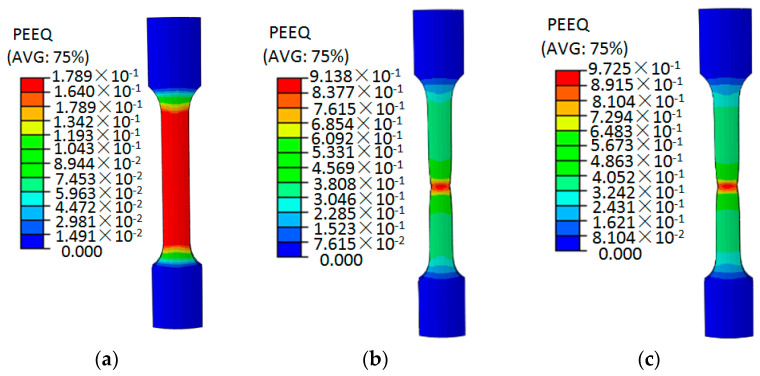
Equivalent plastic strain distribution of 6061 aluminum alloy circular arc specimen at different tensile deformation stages. (**a**) 9.8 mm; (**b**) 29.26 mm; (**c**) 30.45 mm.

**Figure 13 materials-15-03212-f013:**
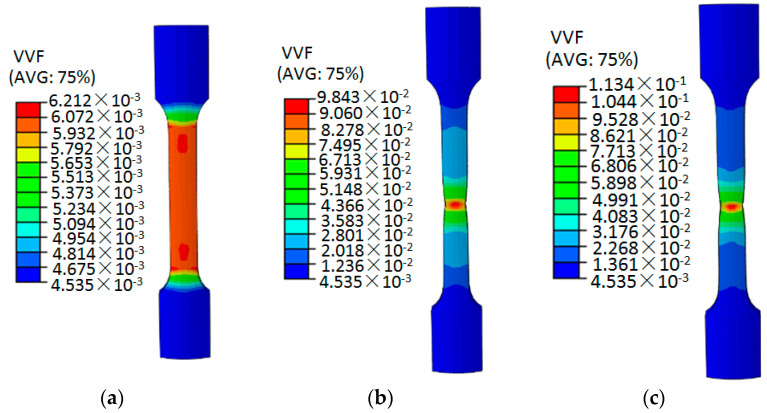
Void volume fraction distribution of 6061 aluminum alloy circular arc specimen at different tensile deformation stages. (**a**) 9.8 mm; (**b**) 29.26 mm; (**c**) 30.45 mm.

**Figure 14 materials-15-03212-f014:**
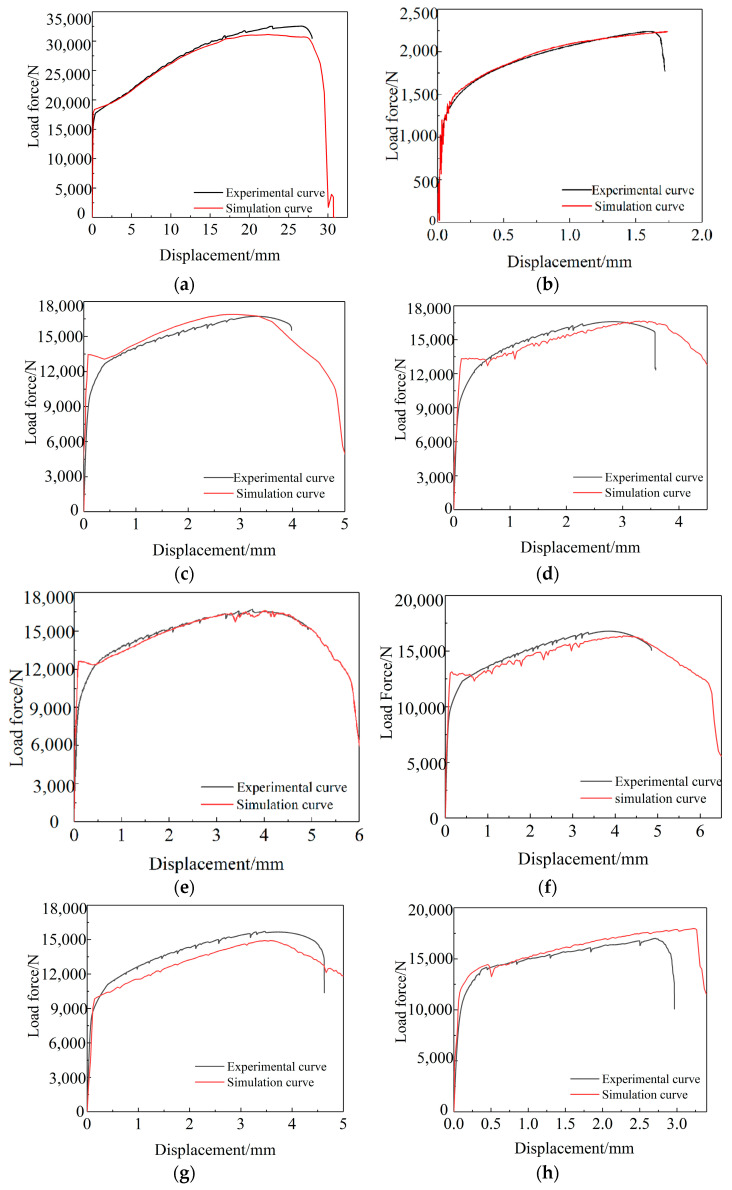
Comparison of experimental and numerical load force-displacement curves by GTN model. (**a**) Arc tension specimen; (**b**) Shear tension specimen; (**c**) R = 5 mm; (**d**) R = 6 mm; (**e**) R = 8 mm; (**f**) R = 11 mm; (**g**) R = 15 mm; (**h**) α = 60°; (**i**) α = 90°; (**j**) α = 120°.

**Table 1 materials-15-03212-t001:** The main component of 6061-T6 aluminum alloy extruded tube (wt.%).

Si	Fe	Cu	Mn	Mg	Cr	Zn	Ti	Al
0.52	0.33	0.21	0.94	0.17	0.16	0.04	0.03	Bal

**Table 2 materials-15-03212-t002:** Mechanical properties of 6061 aluminum alloy tube.

Al	E/MPa	A/%	σ_s_/MPa	σ_b_/MPa	K	*n*	R^2^
6061	65,097	26	152	412.29	592.6	0.446	0.974

**Table 3 materials-15-03212-t003:** Void volume fraction at different tensile stages.

*f*	*f* _0_	*f_F_*
value	0.004535	0.2135

**Table 4 materials-15-03212-t004:** Factors and levels of orthogonal experiments.

Factor Level	Test Factors			
	*ε_N_*	*f_N_*	*f_C_*	*S_N_*
1	0.25	0.005	0.1	0.01
2	0.3	0.01	0.135	0.05
3	0.35	0.02	0.17	0.1
4	0.4	0.04	0.205	0.15

**Table 5 materials-15-03212-t005:** The orthogonal experiment design scheme.

Num	*ε_N_*	*f_N_*	*f_C_*	*S_N_*
1	1	1	1	1
2	1	2	2	2
3	1	3	3	3
4	1	4	4	4
5	2	1	2	3
6	2	2	1	4
7	2	3	4	1
8	2	4	3	2
9	3	1	3	4
10	3	2	4	3
11	3	3	1	2
12	3	4	2	1
13	4	1	4	2
14	4	2	3	1
15	4	3	2	4
16	4	4	1	3

**Table 6 materials-15-03212-t006:** Calculation results and analysis of orthogonal experiment.

Num	*ε_N_*	*f_N_*	*f_C_*	*S_N_*	ε_calculation_	σ_calculation_	*RE_ε_%*	*RE_σ_%*
1	1	1	1	1	0.54357	459.9223	26.57	11.55
2	1	2	2	2	0.54515	455.5453	26.94	10.49
3	1	3	3	3	0.54677	446.1706	27.32	8.22
4	1	4	4	4	0.5024	404.3585	16.99	1.92
5	2	1	2	3	0.54252	459.6153	26.33	11.48
6	2	2	1	4	0.54527	456.7254	26.97	10.78
7	2	3	4	1	0.54613	446.7440	27.17	8.36
8	2	4	3	2	0.54073	425.1145	25.91	3.11
9	3	1	3	4	0.54493	461.4392	26.89	11.92
10	3	2	4	3	0.54375	455.9075	26.62	10.58
11	3	3	1	2	0.54675	446.8711	27.31	8.39
12	3	4	2	1	0.4078	374.3980	5.04	9.19
13	4	1	4	2	0.54346	460.5381	26.55	11.70
14	4	2	3	1	0.54358	456.1281	26.58	10.63
15	4	3	2	4	0.52802	442.9896	22.95	7.45
16	4	4	1	3	0.43651	391.0997	1.64	5.14
*RE_ε_*	*K* _1*j*_	97.82	106.3	82.49	85.36	---	---	---
	*K* _2*j*_	106.3	107.1	81.26	106.7	---	---	---
	*K* _3*j*_	85.86	104.7	106.7	81.91	---	---	---
	*K* _4*j*_	77.72	49.58	97.33	93.8	---	---	---
	*K*_1*j*_/4	24.45	26.58	20.62	21.34	---	---	---
	*K*_2*j*_/4	26.59	26.77	20.31	26.67	---	---	---
	*K*_3*j*_/4	21.46	26.18	26.67	20.47	---	---	---
	*K*_4*j*_/4	19.43	12.39	24.33	23.45	---	---	---
	*R_j_*	7.16	14.38	6.36	6.2	---	---	---
Factor priority		*f_N_*	*ε_N_*	*f_C_*	*S_N_*	Optimal level combination		(*ε_N_*)_4_(*f_N_*)_4_(*f_C_*)_2_(*S_N_*)_3_
*RE_σ_*	*K* _1*j*_	32.18	46.65	21.35	39.73	---	---	---
	*K* _2*j*_	33.73	42.48	38.62	33.69	---	---	---
	*K* _3*j*_	40.08	32.42	33.88	35.42	---	---	---
	*K* _4*j*_	34.92	13.14	32.56	32.07	---	---	---
	*K*_1*j*_/4	8.04	19.36	5.33	9.93	---	---	---
	*K*_2*j*_/4	8.43	10.62	9.65	8.42	---	---	---
	*K*_3*j*_/4	10.02	8.1	8.47	8.85	---	---	---
	*K*_4*j*_/4	8.73	4.84	8.14	8.01	---	---	---
	*R_j_*	1.98	14.52	4.32	1.92	---	---	---
Factor priority		*f_N_*	*f_C_*	*ε_N_*	*S_N_*	Optimal level combination		(*ε_N_*)_4_(*f_N_*)_4_(*f_C_*)_2_(*S_N_*)_3_

**Table 7 materials-15-03212-t007:** GTN parameters of 6061 aluminum alloy.

*S_N_*	*ε_N_*	*f* _0_	*f_N_*	*f_F_*	*f_c_*
0.1	0.4	0.004535	0.04	0.2135	0.1

## Data Availability

All data, models, and code generated or used during the study appear in the submitted article.

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
