# Peer review of "Prediction of Fracture Behavior of 6061 Aluminum Alloy Based on GTN Model"

_materials, 2022, doi:10.3390/ma15093212_

Round 1

Reviewer 1 Report

Dear Authors,

I have read your paper "Prediction of fracture behavior of 6061 aluminum alloy based on GTN model".

It fulfills the aims and scope of the Metals journal. Presented investigations are interesting. My comments and suggestions are listed below.

General remarks:

  • Please add the quantitative results into the abstact. Moreover, used testing methods should be listed here.
  • You have presented 20 references. I suggest to extend this to 25-30. It allos to increase the visibility of your paper in scientific databases.

Introuction:

  • Firstly, it should started by number "1" not "0".
  • Please describe strutures, in which investigated material is used. Moreover, other problems connected with your materials should be described. It will underline the necessity of your investigations.
  • The novelty of your work should be underlined more. It should be clear for readers.

Experimental:

  • I propose to change the name to "Materials and Experimental". You have describe materials too.
  • Table 1 - the source of presented values is unknown. Have you anayzed this composistion? Or values were taken from standard/manufacturer data? It should be marked in the paper. Moroever, I propose to show general mechanical properties of used alloy (yield point, tensile strength and elongation).
  • Presented equations should be supported by relevant references.
  • Figs 5 an 6 - please show bigger pictures.

GTN model and determination of damage parameters:

  • Please support equations with relevant references. Also units should be presented.
  • Figs 7 and 8 - please show bigger, moreover the scale bar shoud be clearly visible.
  • Tables 3 and 4 - please describe why these values were used. It should be clear for all readers.
  • Figs 9, 13 - it is impossible to read information from so small picture.
  • The modelling shoud be described in details (boundary conditions, no of finite elements and their shape).
  • Could you show the real macrophoto of your specimens.

I cannot find any scientific discussion in your paper. Now, it looks like technical raport. You shoud discuss and compare your results with other scientific papers. It allows to underline the biggest advantages from your investigations. Without good scientific discussion, paper cannot be published as a scientific work.

Conclusions:

  • Please support conclusions by quantitative results.

Author Response

Thank you for your careful review. We really appreciate your efforts in reviewing our manuscript during this unprecedented and challenging time. We wish good health to you, your family, and community. Your careful review has helped to make our study clearer and more comprehensive.

Reviewer 2 Report

Authors:

Fengjuan Ding, Tengjiao Hong, Youlin Xu and Xiangdong Jia

Title of the manuscript:

Prediction of fracture behavior of 6061 aluminum alloy based on GTN model

Manuscript ID: materials-1661850

The authors work with the GTN damage model for 6061 aluminum alloy implemented in Abacus FEM software. The presented experimental work is related to mechanical testing (axial tensile tests) of 6061 aluminum arc samples and to identifying optimum model parameters (SEM and orthogonal test method). The authors claim that the GTN damage model can effectively predict the fracture failure of 6061 aluminum alloy. Research of aluminum alloys is of high importance. However, the weak point of the manuscript is a lack of clarity. If the authors succeed to improve clarity of the manuscript, then their work is worth publishing in the journal. The following points should be addressed:

  1. Meaning of GTN is not introduced in the abstract and the introduction.
  2. Line 41 wrong grammar: “Have a significant impact.” + “… model to studied”
  3. Line 45 wrong grammar: “model to investigated”
  4. Line 46 wrong grammar: “when the simulated load-displacement curves of small punch testing specimens accordant with the experimental results.” No verb in sentence.
  5. Line 50, Style: “Bergo et al.[5] discussed the effect of the material characteristic length on the ductile damage and fracture behavior and on the mesh sensitivity of the results, the numerical research showed that simulation results obtained in all stages of the ductile fracture process, including void growth, fracture initiation by coalescence and crack propagation all the way to a fully fractured specimen, were mesh independent for a certain mesh size ratio related to the material characteristic length, provided the non-local integral was evaluated on the current configuration.” The sentence is too long.
  6. Line 54, FE, meaning not introduced.
  7. Line 57 wrong grammar.
  8. Figure 4. The samples are not flat. It would be nice to show some other complementary views where it is clear how the samples are bent in 3D.
  9. Equations in the manuscript: Some parameters are not properly introduced, or their definition is missing in the text. Some symbols/parameters are introduced more than once in the manuscript.
  10. Figure 6: It seems that a simple linear fit would be better that the one presented by the authors.
  11. Table 2: Some parameters are not explained.
  12. “…assumption that the matrix material is incompressible…” It seems too strong assumption. Poisson ratio for incompressible materials is close to 0,5 whereas Poisson ratio of aluminium is about 0,33. Can the authors comment on this?
  13. Line 193: the authors likely mean formula 9
  14. Line 196: I recommend that the authors use a better equation editor.
  15. Line 190: What is meant by static water component?
  16. Table 3, 4: It would be nice if the symbols in the table have their names there or at least a reference to Equation where they are introduced.
  17. Line 236: Why to use imperative form; better: We modified the GTN model...
  18. Line 237: “…the error between the simulation curve and the test curve” The meaning of the sentence is not clear.
  19. Line 238: What is meant by the best GTN damage parameters? The best GTN damage parameters are best from which point of view?
  20. Line 246: “Needleman[17] A pointed” It is better to remove “A”
  21. Some other unclarities or grammar mistakes are in the following lines: 256, 265, 284-285, 296, 303-304, 306, 307, 370-374, 378, 387, 392, 396-397,
  22. Table 6: the table title refers to an experiment but some of the parameters in the table are labelled "calculation". Can the authors check this little discrepancy?
  23. Figure 13: Can the authors explain why the displacement in Fig. 13a is ten times larger than in the other cases (b, c, d…)?

Author Response

(The authors gave the same response as above.)

Reviewer 3 Report

This paper deals with the GTN fracture model for 6061 aluminum alloy. This paper lacks scientific discussions about the fracture model. Consequently, I did not recommend this paper to be published.

1. This paper works with the GTN fracture model. GTN model means the fracture model of Gurson-Tvergaard-Needleman. The description of GTN model should be addressed.

2. There were some errors in English. in line 97,

"According to GB/T228-2010《Metal Material Room Temperature Tensile Test Method》, cut the arc specimen, shear specimen, arc notch specimen and V-notch specimen along the axial direction of 6061 aluminum alloy tube, as shown in Figure 1 to Figure 4, heat treatment is performed on different types of aluminum alloy specimens (heating temperature was 560℃, holding time was 4h, and then were cooled in water )"

What is the subject of the first sentence? This paragraph is too long to understand clearly. What is the meaning of "shown in Figure 1 to Figure 4"? 

Also, in  (heating temperature was 560℃, holding time was 4h, and then were cooled in water ) why did you use ( ) ? 

In Line 266, 'and' should be used instead of 'And".

3. How did you cut the specimen? In line 98, you just mentioned cut the arc specimen. Is it necessary to flatten the specimen after cut the specimen from the pipe?

4. In line 151, a simple power-law such as sigma=K e^n was employed. As shown in Fig. 6, the stress-strain curve showed large difference between the simulation results and the experimental results. In Figure 6, you just started from the yield stress. In this case, Swift model sigma = K (eo + ep)^n shows better results. 

Also, the stress-strain relationship is nearly linear as shown in Fig. 6. In that case, n will be nearly 1.

Tha main problem of this paper is in Figure 9, 10, and 13. When you employed the stress-strain relationship in Eq. (4) and Table 2, the simulation results should follow the stress-strain relationship in Fig. 4. However, in the simulation results shown in FIgure 9, 10, and 13, the stress-strain curve of the simulation result followed the experimental result. How do you think about this one?

This is a critical problem.  

I think you just write down the fitting results in Table 2. In the simulation you just employed the stress-strain with table data of experiments. Is it right?

5. In GTN model, measuring the void volume fraction is very important. However, information about the void volume fraction is very small. For example, how did you take SEM images? In what directions? How many SEM images were taken to get average values?

6. The results of GTN parameter is summarized in Table 7. epsilon_n and f_n is the largest value in Table 4. The design of the Table 7 is wrong. What happen if e_n and f_n is larger than the test factors?

7. What is the meaning of "The driving force of great need" in line 378?

8. You mentioned your model does not follow the experimental results with low-stress triaxiality. What is "low-stress triaxiality"? All of your experiments are tensile tests. What are experiments with "low-stress triaxiality"?

Author Response

(The authors gave the same response as above.)

Round 2

Reviewer 1 Report

Dear Authors,

Your paper has been improved a lot. Your efforts are appipiate. I have one minor remark, which was not included in revised paper.

I propose to show general mechanical properties of used alloy (yield point, tensile strength and elongation).

Author Response

(The authors gave the same response as above.)

Reviewer 2 Report

The revised manuscript is improved. However, there are still typos, grammatical and stylistic errors, e.g. lines 65, 70, 106, 102 (while->whereas), 104(unexplained abbreviation EEM), 84-89(too long sentence), etc.  I recommend the authors to check the manuscript more carefully.

Author Response

(The authors gave the same response as above.)

Reviewer 3 Report

The paper was well revised. However, some errors should be corrected. Please read carefully and correct some errors.

The flow stress model should be rigorous. In the simulation, the flow stress from the experimental results was used directly. sigma=K e^n was employed in the fracture modeling. Please mention that.

Author Response

(The authors gave the same response as above.)
